# 'I don't want anyone to know': Experiences of obtaining access to HIV testing by Eastern European, non-European Union sex workers in Amsterdam, the Netherlands

**Anna Tokar**[1]*, **Jacob Osborne**[2], **Robbert Hengeveld**[2], **Jeffrey V. Lazarus**[1], **Jacqueline E. W. Broerse**[2]

**1** Barcelona Institute for Global Health (ISGlobal), Hospital Clínic–University of Barcelona, Barcelona, Spain, **2** Athena Institute, Faculty of Science, Vrije Universiteit Amsterdam, and Amsterdam Public Health Research Institute (APH), Amsterdam, the Netherlands

* anna.tokar@isglobal.org

**Data Availability Statement:** The datasets generated and/or analysed during the current study are not publicly available due to sensitivity of the

## Abstract

Historically, the Netherlands has hosted a large number of migrant sex workers. Since sex work is considered a legal profession it might serve as an example of better access to health services, including HIV testing, at least for those working within the legal framework. However, migrant sex workers, especially non-European Union (EU) nationals, might not be eligible to register for official employment and thus face obstacles in obtaining access to health services, becoming essentially invisible. This study examined context-specific vulnerabilities of migrant female sex workers (FSWs) from Belarus, Moldova, Russia and Ukraine, whether and how they have access to HIV testing compared to other EE, non-EU migrant FSWs in Amsterdam in the Netherlands. We conducted a multi-stakeholder perspective study from November 2015 to September 2017 in Amsterdam. The study comprised 1) semi-structured interviews with key stakeholders (N = 19); 2) in-depth interviews with Eastern European, non-EU migrant FSWs (N = 5) and field observations of the escort agency working with them; and 3) in-depth interviews with key stakeholders (N = 12). We found six key barriers to HIV testing: 1) migration and sex-work policies; 2) stigma, including self-stigmatization; 3) lack of trust in healthcare providers or social workers; 4) low levels of Dutch or English languages; 5) negative experience in accessing healthcare services in the home country; and 6) low perceived risk and HIV-related knowledge. Having a family and children, social support and working at the licensed sex-work venues might facilitate HIV testing. However, Internet-based sex workers remain invisible in the sex-work industry. Our findings indicate the importance of addressing women's diverse experiences, shaped by intrapersonal, interpersonal, community, network and policy-level factors, with stigma being at the core. We call for the scaling up of outreach interventions focusing on FSWs and, in particular, migrant FSWs working online.

topic. Although we did not retain any personal identifiers of the participants, all the transcripts will be further anonymised in order to guarantee participants' privacy.' Requests can be directed to ISGlobal, Hospital Clinic, University of Barcelona (just as kind recommendation better to write in Spanish). Ethics Committee emails: CEIC@clinic. cat; proceic@clinic.cat.

**Funding:** This study was financed by the TransGlobal Health Program as a part of the Erasmus Mundus Joint Doctorate Programme.

**Competing interests:** The authors have declared that no competing interests exist.

## Introduction

Current debates on sex work are highly polarized: while opponents portray sex work as exploitative and coercive, and never a free choice, proponents suggest recognizing sex workers' agency and that denying the opportunity to engage in sex work is a direct violation of civil rights [1]. Consequently, these arguments have entered the public health agenda, where both camps sought to provide services either through the 'rescuing industry' (anti-trafficking organizations) or through sex workers' rights organizations [1, 2]. Victimization and rescuing women for their own good goes back to the Jane Addams' construct of 'White Slavery' in the 1950s, which viewed sex workers as feeble-minded pathological deviants [3, 4], and the most at-risk populations and bridging populations at the peak of the AIDS era [2, 5]. Overall, EU policy discourse equated trafficking with sex work, making it impossible to separate laws that regard non-EU migrant sex workers as potentially trafficked victims [6].

The Netherlands is the only EU country where sex work is considered a legal profession. Even so, since the legalization of sex work in 2000 and introduction of stricter measures to combat human trafficking (2009) there have been debates on the effects of legalizing sex work. In practice, these laws have led to a growing illegal sector, pushing individuals out of the officially recognized and registered sex-work venues within the city councils, multiplying their vulnerabilities and limiting their access to health services [7, 8]. Moreover, taking into account the historically high representation of migrant sex workers in the country (up to 60–75%) [9], such measures produced contradictory effects for migrants, especially for non-EU nationals, as they are ineligible to register for official employment unless they have valid work and a residence permit [7, 8].

### HIV and AIDS and migrant sex workers in the Netherlands

The prevalence and incidence of HIV in the general population in the Netherlands is relatively low, with an estimated 19,582 people living with HIV (PLHIV), and 749 new cases diagnosed in 2017 [10]. Overall, HIV prevalence among sex workers is also reported to be low in the Netherlands, with exception of transgender and sex workers who use drugs [11, 12]. Although there is no robust evidence, the proportion of sex workers and migrants among all PLHIV seems to be somewhat higher than in the general population. Migrants accounted for 63% of all PLHIV who acquired HIV through heterosexual transmission in 2017 [10]. A cross-sectional study undertaken by van Veen et al. showed an HIV prevalence of 1.5% among female sex workers (FSWs) who did not inject drugs[11]. Furthermore, de Coul et al. showed that 351 female sex workers were living with HIV of the estimated number of 25,000 female sex workers across the Netherlands–an HIV prevalence of 1.6% [13].

It appears that sex workers living with HIV are more often unaware of their status than PLHIV in the general population [14]. Specifically, 74% (23/31) of all sex workers living with HIV were unaware of their infection in the study undertaken by van Veen et al.[11], compared to an estimated 10% in the general population. (In total, 20,844 individuals, 90% of the total number estimated to be living with HIV, had been diagnosed, linked to care[10].) De Coul et al. estimated that around 66% of all female sex workers in the Netherlands might be unaware of their HIV-positive status–and that this proportion did not change between 2007 and 2012 [13]. As was suggested by van Veen et al. (2010)[11], awareness of HIV status ought to be enhanced among sex workers in the Netherlands by improving HIV and sexually transmitted infection (STI) testing facilities and access to care. Surprisingly, sex workers do not seem to be an identified risk group for HIV infection in the Netherlands–national HIV reports stratify HIV cases only for men who have sex with men (MSM), people who inject drugs (PWID) and migrants, but not for sex workers [10].

These studies strongly suggest that migrant sex workers from non-EU countries are a significant HIV-risk group in the Netherlands. The combination of being a sex worker and a migrant together with being unable to work legally is likely to lead to barriers in obtaining access to health services. Those barriers could contribute to a low uptake of HIV-testing services, and therefore to being unaware of HIV status. This could have negative health consequences at an individual level as well as among the population group.

### Eastern European, non-EU migrant female sex workers

An estimated 47% of all female sex workers in the EU are migrants, of whom 70% are from Eastern European (EE), non-EU European countries and Central Asia [9]. Since 1991, after the collapse of the Soviet Union, an increased number of male and female sex workers from Central and Eastern European (CEE) countries were reported in many EU countries [15–18]. A significant increase in attendance of EE female migrants at clinics for sexually transmitted infections (STIs) was reported in several EU countries [19, 20]. Further enlargement of the EU will most likely lead to more EE migration to Western Europe [21]. This is particularly the case for candidate member states. Moldova and Ukraine have signed an Association Agreement, which is a preparatory stage for EU membership and/or and simplification of visa procedures for their citizens. This may lead to more migration from Ukraine and Moldova, comparable to the population movements from the group of Central and Eastern European (CEE) countries that entered the EU in 2004 (known as the EU8) and 2007 (the EU2)[22].

Eastern Europe and Central Asia are the only regions in the world where new HIV infections continue to rise rapidly, with 103,438 new cases diagnosed in 2016 [23]. In addition, prohibitive administrative (Statute 181/1) [24] and criminal [Statutes 130, 302, 303][25] laws may fuel the stigmatization of sex workers as well as lead to violations of their human rights. Several recent public debates on the development of the first non-discriminatory sex-work policies in Ukraine were called off in response to profound resistance from government and civil society to recognizing sex work as a profession. As other research studies have demonstrated, the prohibition of sex work can lead to stigma and discrimination, social exclusion, unsafe working conditions, poor occupational health, low self-esteem, and restrictions on housing, travel and parenting [26–28]. Thus, even though testing is available in Ukraine, stigma and criminalization may constrain accessibility and uptake, or may affect the disclosure of HIV-positive status, resulting in the forced invisibility of female sex workers and their needs.

Although access to HIV-testing services is considered to be relatively good in the Netherlands, to our knowledge there has been no study that has examined this among a population group for whom this access can be significantly different.

Moreover, although a majority of sex workers in Netherlands are migrants, with EE nationals estimated to be the largest group [9], there is limited information on EE, non-EU nationals. In this study we focused specifically on this population, aiming to 'unpack' context-specific vulnerabilities of migrant female sex workers from EE, non-EU countries (Belarus, Moldova, Russia and Ukraine), and to describe whether and how these affect their access to HIV testing in Amsterdam in the Netherlands.

## Methods

In this study, we use the Joint United Nations Programme on HIV/AIDs (UNAIDS) definition of sex work: 'Female, male and transgender adults and young people (over 18 years of age), who receive money or goods in exchange for sexual services, either regularly or occasionally' [29]. This study focuses on girls and women, not only because they make up the biggest proportion of all sex workers in Europe (up to 87%) [9], but also because they are more likely to

be possible victims of human trafficking in the Netherlands, characterized by sexual exploitation [30]. The EU and EE, non-EU member states are determined in accordance with Schengen Visa Countries List [31]. We define as 'EE, non-EU migrants' all persons who were born or spent formative years in Belarus, Moldova, Russia or Ukraine.

## 2.1 Study design and setting

We conducted a multi-stakeholder perspective study from November 2015 to September 2017 among key stakeholders (N = 36) in Amsterdam (see *Table 1*).

The study comprised three phases: 1) semi-structured interviews (SSIs) with key stakeholders who either worked with sex workers and/or migrants (N = 19); 2) in-depth narrative interviews (IDIs) with EE, non-EU migrant female sex workers (N = 5) and field observations of the escort agency working with them in Amsterdam; and 3) in-depth narrative interviews with key stakeholders (N = 12) (Fig 1).

## 2.2 Study population and sampling

**Phase 1: Exploring local context.** In order to enter the field as well as to explore how HIV testing among migrant sex workers is provided 'on the ground' we conducted semi-structured interviews (SSIs) with participants, who worked with sex workers, migrants or migrant sex workers. Since we aimed to collect information on predefined topics in this first phase of the pro, we conducted semi-structured interviews, in which a researcher set the agenda in terms of topics covered, but the respondent determined the kinds of information obtained as well as the relative importance of this information [32].

The participants we recruited purposively in order to include more informative persons with regard to testing, sex workers, migrant sex workers and the latter's access to health services. Specifically, we included health workers, social nurses and social workers, researchers focusing on issues of sex work and migration, NGOs' representatives working with sex workers/migrant sex workers and self-organized organizations of sex workers, a journalist/representative of the company producing female condoms and a city council representative. All interviews were conducted by AT or RH in English or Dutch and on average lasted 40–100 minutes.

**Phase 2: Investigating everyday routine of EE, non-EU migrant sex worker in Amsterdam.** In Phase 2 we used in-depth, narrative interviews as this method allowed respondents to have enough time to develop their own accounts of the issues important to them, while interviewer served rather as a facilitator of this process. Respondents determined the conversation flow [32].

Respondents were recruited using a convenience sampling strategy via web pages, where sex work is advertised in Russian and English for Amsterdam and via the key NGOs providing services for female sex workers and migrant female sex workers in Amsterdam. Online recruitment was via electronic messages, describing in brief the purpose of the study, contact information and the compensation for their time (a mobile top-up of 25 EUR). Key NGOs to contact for recruitment purposes were identified using 'Services4sexworkers' (which offers a directory of services available for sex workers in 25 European countries) and The Global Network of Sex Work Projects (NSWP) websites, which aim to uphold the voice of sex workers globally and connect regional networks advocating for their rights. The search via this website for Amsterdam was conducted one month prior to the field work, to obtain the most recent NGO contact information. The European Network for HIV/STI Prevention and Health Promotion among Migrant Sex Workers (TAMPEP), which is based in Amsterdam, was also contacted. We were also guided by the information collected during Stage 1 of the Project. All

**Table 1. List of participants.**

| Participant, N | Position | Institution |
|---|---|---|
| Phase 1: Exploring local context, semi-structured interviews with key stakeholders, (N = 19) | | |
| 1 | Social nurse (specialized in SWs) | Municipal health service |
| 2 | Social nurse (specialized in SWs) | Municipal health service |
| 3 | Psychologist | |
| 4 | Social nurse (specialized in SWs) | Municipal health service |
| 5 | Physician | |
| 6 | Physician, researcher | Municipal health service, Academia |
| 7 | Social nurse, director of non-government organization (NGO) | Municipal health service, non-governmental organization working with women |
| 8 | Healthcare provider | Municipal health service |
| 9 | Senior Program officer for HIV/STI prevention and health promotion among SWs | Government organization |
| 10 | Programme manager prostitution | Government organization |
| 11 | Social worker | Governmental social services |
| 12 | HIV counsellor & tester | Non-government organization |
| 13 | Director, SW activist | Non-government organization |
| 14 | Director | Non-government organization |
| 15 | Director | Non-government organization |
| 16 | Researcher | Academia |
| 17 | Researcher | Academia |
| 18 | Journalist | Media |
| 19 | City council representative | Government organization |
| Phase 2: Investigating everyday routine of EE, non-EU migrant sex workers in Amsterdam, in-depth (narrative) interviews with EE, non-EU FSWs, escort agency (N = 5) & field observations of the escort agency | | |
| 1 | EE, non-EU FSW, with Dutch residency | |
| 2 | EE, non-EU FSW, with Dutch residency | |
| 3 | EE, non-EU FSW working on tourist visa | |
| 4 | EE, non-EU FSW working on tourist visa | |
| 5 | Owner of escort agency | |
| Phase 3: Finding ways forward, in-depth (narrative) interviews with stakeholders, (N = 12) | | |
| 1 | Social nurse (specialized in SWs) | Municipal health service |

**Table 1.** (Continued)

| Participant, N | Position | Institution |
|---|---|---|
| 2 | Project manager | Government organization |
| 3 | Coordinator | Non-government organization |
| 4 | Coordinator | Non-government organization |
| 5 | Advisor in the area of sex work and migration | Government structure |
| 6 | Street-based sex worker, activist | Non-government organization |
| 7 | Policy officer | Non-government organization |
| 8 | Social nurse (specialized SWs) | Municipal health service |
| 9 | Researcher | Academia |
| 10 | Training specialist | Non-government organization |
| 11 | Programme Associate | Non-government organization |
| 12 | Police officer | Government organization |

NGOs were first contacted by email, and then, if there was no response, by phone to present the key objectives of the study and to propose collaboration on the study. Moreover, NGO staff were asked to disseminate the flyers (in Russian and English), with brief and clear research objectives and procedures. Before they were released for distribution, the flyers were reviewed by a sex-work activist and two project managers working in HIV-prevention projects among key affected populations. All interviews were conducted by AT via phone or online call (WhatsApp, Viber or Skype) in Russian, no audio-recording was permitted so notes were taken. Interviews lasted for 60–180 minutes.

**Phase 1: SSI interviews with key stakeholders (N=19)**

**Objectives:**

➢ to understand how HIV testing is delivered "on the ground"
➢ to identify ways to recruit EE, non-EU migrant FSWs
➢ to inform guideline for IDI with EE, non-EU migrant FSWs and stakeholders

**Phase 2: IDI with EE, non-EU migrant FSWs (N=5) & field observations**

**Objectives:**

➢ to collect views and experience of HIV testing of EE, non-EU migrant FSWs
➢ to collect views how to improve HIV testing
➢ to inform guideline for IDI with stakeholders

**Phase 3: IDI interviews with stakeholders (N=12)**

**Objectives:**

➢ to collect views of stakeholders towards experiences of HIV testing among EE, non-EU migrant FSWs
➢ to collect views how to improve HIV testing among EE, non-EU migrant FSWs

**Fig 1. Project phases.**

Over a 12-week period (June–October 2016) we observed the work of an escort agency working with EE, non-EU female sex workers in the Netherlands. Thus, for eight weeks we had weekly meetings (either as online chats or via Skype or Viber call) with the owner of this escort agency to discuss the current situation and how work is organized, including healthcare issues and how access to health services is assured during the female sex workers' journey. We also asked the participant to contact us if there were relevant issues. For instance, we were approached by the owner of this escort agency when one of the women was deported or when the agency received an email from the City Council of Amsterdam demanding them to stop working in Amsterdam and in the Netherlands. Moreover, for four weeks (September–October 2016) we visited its website daily to collect advertisements of female sex workers (N = 8) as well as clients' feedback posted on the escort agency's website (N = 42). Screen shots of all webpages were taken, dated, and archived for analysis. We extracted the data using a standard form (S1 Appendix) from each individual advertisement to assess information about services provided, health and safety information, prices, work location (in-call or out-call) and working hours.

**Phase 3: Finding ways forward.** Study participants were recruited purposively; our choices were informed by the data received during Phase 1 and 2 of the study. All interviews were conducted by AT, in English, and lasted between one and two hours.

## 2.3 Data collection, management and analysis

*Semi-structured interviews with key stakeholders* were conducted on the basis of the interview guide (S2 Appendix), which included six topics: sex work, migration and HIV-testing policies and laws, provision of HIV-testing services among migrant sex workers, perceptions of health providers, perceptions of HIV testing among migrant female sex workers, and identification of the key stakeholders.

*In-depth interviews were very loosely structured, and usually covered topics such as*: migration patterns in the Netherlands, knowledge and experience of HIV testing, facilitators and barriers to HIV testing, HIV-testing approaches and modalities, and suggestions on how to improve the uptake of HIV testing. Yet, respondents determined the topics covered, which allowed them to "tell their story". We asked three independent experts (a sex worker and activist, and two project managers working in HIV-prevention projects among key affected populations) to review the guidelines (S3 and S4 Appendixs). The research team carefully considered their comments and suggestions, which were incorporated as appropriate. Data gathered and analysed informed the next interviews.

After each interview a debriefing form was completed and data was discussed within the research team. All notes were transcribed verbatim. Random fragments were sampled for quality checks and imported and analysed using the qualitative software ATLAS.ti 8. We developed a codebook to examine barriers to and facilitators of HIV testing among EE, non-EU migrant female sex workers using a socio-ecological framework (SEM) as well as our research questions. SEM illustrates multiple associations between different factors of micro, meso and macro levels (including intrapersonal, interpersonal, community, network and policy) and how they relate to health outcomes [33] and inequities and disparities [34–36].

Two researchers (AT and JO) used the conceptual framework: barriers and facilitators of HIV testing amongst FSWs (retrieved from: Tokar et al., 2018 [36]) to code data independently. Based on conversations between researchers the coding framework was modified, codes were reviewed, new codes emerging from the data were included and codes were categorized thematically (S5 Appendix). Both researchers aimed to structure data taking account of significant meanings in each text as well as the relationships between certain parts of the texts.

When interviews were conducted in English with non-native speakers, quotes cited in the text were slightly grammatically corrected to improve their readability.

While 'she'/ 'he' pronouns were employed irrespectively of the sex and gender of research participants, we used 'she' consistently when talking about a female sex worker.

A more detailed descriptions and reflections on methodological aspects of this study, including sampling, data collection and ethical issues, could be found in another article, which we have written.

### 2.4 Ethical considerations

This study has been reviewed by Institutional Review Board of both Vrije Universiteit, Amsterdam, the Netherlands and ISGlobal, Hospital Clinic, University of Barcelona, Barcelona,Spain. As this study was deemed 'non-invasive' by Dutch law, and all participants were over 18 years of age, we did not require approval from a formal medical ethical committee [37]. The researchers adhered to the national Code of Ethics for Research in the Social and Behavioural Sciences involving Human Participants [38]. Verbal informed consent was collected from each participant. All participants were given a detailed description of the study. Depending on the language preference, the consent form was administrated either in Russian or English. All respondents gave informed consent and all the interviews were anonymous and confidential. Verbal informed consent was obtained from each participant when participants agreed to an audio-recorded interview. In cases, when the interview was not audio-recorded, written informed consent was obtained, including text messages via Skype or Viber. In two cases when the interview was conducted via mobile phone with EE, non-EU female sex workers and permission to record was not granted, we could not record informed consent.

Study participants could choose a convenient time and location for the interview. For privacy reasons, all interviews with female sex workers were conducted either by phone or online (via WhatsApp, Skype or Viber). Separate phone number, Skype and Viber accounts were created and used during the field research to recruit and maintain contact with participants. Interviews with stakeholders were conducted in private settings, usually in a separate room in their office; several interviews took place in a café, where we chose location offering maximum possible privacy (ranging from empty because of the cold-weather terrace, or the second floor). No personal data was collected and all interviews were anonymous. All the data was kept safely in a password-protected office computer to which only researchers involved in the project have access. All female sex workers were offered a digital code to top up their mobile phone (25 EUR) as compensation for time spent.

## Results

### Context of sex work among EE, non-EU migrant women in Amsterdam

Overall, most of the study participants suggested that because of legal restrictions it is very difficult for non-EU residents without a valid residence and work permit to be employed as a sex worker in Amsterdam, but several strategies for how to overcome this barrier were reported. Women either had to look for opportunities to get documents through legally recognized pathways (family reunification, student visa, or legal job other than sex work) or they advertised sexual services through specialized web platforms in advance, and then came as tourists for two- to three-week periods, moving across several cities or even countries, and providing services to a pre-agreed list of clients. In the latter case, women would try to escape the attention of public authorities, including health workers and social workers. This could make them invisible, unless they were detained by the Dutch police or migration services. Even though

some NGOs and health facilities have online outreach programmes aimed at online sex workers, it is unknown how effective these are.

## Micro-level factors

**a) Socio-demographic characteristics.** Most of the participants interviewed agreed that the majority of female sex workers are 20–35 years of age, with more young women in the escort services and working online as women below the age of 21 years are not eligible to engage in sex work in Amsterdam. Analysis of the escort agency's online advertisements showed that all women said they were under 25 years of age. The agency participating in the study only invited women under the age of 30 for a job. Yet, several participants reported that age was often misreported as women tried to attract more clients:

> *Yes, that's interesting, because they are not that young as you expect. [. . .] it's all just a lie, the advertisement. That's something I really like–the age is a total lie, it's just to attract the customer.* (Police officer)

For example, one woman described how she had to change her lifestyle in order 'to look like 20s' as most clients seek to dominate, and use their power to get extra sexual services for free:

> *I am 37, but I look like 20 years old. I am on a diet, I don't eat meat, only fish and I work out. No sugar, I eat ice cream only once a year! Clients would never go with me, if they knew that I am 37, never. They all look for young girls, because they can force them to do what they like and not pay for it. Clients ask how old are you? I say 22, they agree, they pick me. They pay money, we go upstairs and when we enter the room, I tell him to do this and that. I am the Boss, not he. I do what I want, and then, they say 'you are not 22, you are older'.* (Female sex worker)

Most of the participants linked youth with a lack of HIV-related knowledge, knowledge of local policies and low self-perceived risk. Moreover, younger women were more likely to work online as escort girls or paid dates, and not perceive themselves as sex workers. Thus, women in the escort services or doing paid dates would not apply for centres using the words 'prostitution' or 'sex workers', and they would not report their involvement in sex work during health checks:

> *Yes, yes, especially the young women speak that language, like car dates, also from Ukraine or Russia they all speak that way because of this [showing smart phone]. . . . and most of young people we see, migrants are very surprised that what they do is maybe prostitution, they don't even know that they are in prostitution, they do not consider this as prostitution as I said they have car dates or paid dates.* (City council representative)

> *There is a group who does sex work, but they do not think they do it. They think 'I am not a sex worker I just receive money for sex'. So, if I am telling them, 'oh do you want to. . . to go to. . . sex workers' centre?' I am not addressing them. No, because they don't feel. . . like there is a place for them there. . .* (Social nurse)

The majority of stakeholders interviewed identified language as an important mediator which could either facilitate or impede women´s navigation through the local context. While knowledge of English and Dutch played a principal role in finding 'your way around', it was poor among EE female sex workers. Moreover, health professionals could not distinguish between different '*Slavic languages*', as according to one social nurse '*Slavic languages look similar*'.

*Interviewer***:** *Why have you mentioned language?*

*Respondent***:** *Well, because it's quite bureaucratic here in Holland, they need papers and stamps, [. . .] and to find your way around, it can be quite difficult. . .if you don't speak the language you can be easily taken advantage of. So, I think especially in sex work, where it is quite difficult to understand tax paying, and the permits. . .even for me it's not very easy to understand. . .* (Police officer)

Moreover, in some cases female sex workers fully relied on a third person (operator) whom they hire for security and language purposes. Thus, operators might fully mediate communication between them and the clients. For example, a client described this situation as follows:

*Now, what went completely wrong–the communication. E. doesn't speak English well. . .50% of the time, we could understand each other and the other 50% of the time, we used Google Translate (not perfect, but okay), and so we could communicate: 7 out of 10. The reason for the 0/10 score, is because the agency/operator promised me and told me things that aren't true. . .*(Retrieved from the website of the escort agency, section 'Clients´ feedback')

According to some stakeholders, key government and non-government organizations providing services for sex workers try to have some part of their information materials translated into Russian, but it was not seen as a key priority as EE, non-EU migrant sex workers are not the largest group in the Netherlands.

**a) Risk behaviours and risk perceptions and knowledge.** Drug use was rarely mentioned and participants talked about the past: '. . .*we hardly have cases of drug addicts among sex workers. It used to be in the 70s. . .*'. Drug use was linked with low perceived risk, illegal employment and working outdoors. One health worker mentioned hearing 'rumours' about the high level of drug use among EE sex workers.

Alcohol consumption was frequent among sex workers, serving as a common strategy to de-stress after work; in some cases, sex workers were supposed to push clients to buy expensive alcohol from a club and then drink it together. Participants believed that alcohol increases risky sexual behaviour while reducing women's perception of risk. For example, a woman working in a club described one situation when, after drinking with a client, she did not notice that during sexual intercourse he removed his condom. For fear of pregnancy and under pressure of her co-workers, whom she informed about the incident, the woman applied to the Prostitution Information Centre in order '*to test and to have a pill'.*

Only one female sex worker interviewed reported insisting her clients use a condom; according to the owner of the escort agency, if clients pay more they do not use condoms and it is typical not to use condoms during oral intercourse. Moreover, according to the analysis of the online advertisements, the majority of women provided oral sex without a condom or would allow the client to ejaculate into her face/mouth for an extra 50EUR. During one of the follow-up interviews the respondent reported having a sore throat, which she believed could be a symptom of an STI, yet because of fear of being recognized as a sex worker, who is providing sexual services on a tourist visa and has little knowledge of the local rules, she did not visit any health facility and continued her 'tour'.

*I do not go in long tours, I can do it [testing] at home . . . and I don't want to draw attention . . . you know, they [health workers] can ask questions . . . I am on a tour now. . . I am ill, I have bad tonsillitis, you know?. . . and I need antibiotics, but without prescription they do not*

*sell it. . .I asked. . .and I even do not know how they go to doctors here and where I need to go, etcs. . . .You know?* (Female sex worker)

Those who came as tourists avoided using any medical services even when they had pre-paid tourist medical insurance and even when they were feeling sick as they were afraid to be identified as a sex worker by the police. Self-medication which was brought from their home country was the most common way to deal with symptoms.

Moreover, according to the analysis of the clients' feedback webpage of the escort agency, almost all women were willing to not use condom during vaginal or anal intercourse, if they were paid extra money. Several misconceptions were reported by female sex workers, which they had heard from their peers. So, female sex workers reported taking a shower before sexual intercourse or having sexual intercourse in a Jacuzzi to reduce the risk of STI transmission, and that sexual intercourse with married clients was perceived to be safer than with single clients.

Several participants discussed the importance of education. A low educational level was linked to low perceived risk and more risky behaviour. The educational component in Russian was a welcomed improvement to the programme:

*You should look at the level of education. Do they know what is important when you are doing sex work, and do they know that it is important to get tested, to get treatment, when necessary?* (Social nurse)

**b) Previous experience.** All women interviewed reported having *'my own doctor'* in their home country. Usually, this refers to a doctor, a trusted person, who is paid out of pocket, able to maintain confidentiality and does not put women into the official records. Moreover, this practice was also reported by stakeholders as a common strategy to reduce costs for health insurance among both EU and non-EU sex workers. Negative experience of accessing health services in the home country could also be a barrier when seeking health care in the country of destination. One woman described her experience in a small city in her home country where she could not go to the local AIDS clinic because of the fear that *'someone will see me entering this building'*. The AIDS clinic was seen in the local community as a place *'for drug addicts and gays'*, and thus, she had to find *'my own doctor'*, whom she paid out of pocket and whom she trusted.

As the police were a source of blackmail, extortion, physical and psychological violence in their home countries, women extended their fear of the police to the country of destination. The interviewed women who travelled as tourists and worked illegally all feared that health professionals would report their illegal work status to the police, which could result in deportation. This fear was also fuelled by the manager of the escort agency, who shared information with women about two cases of 'deportation, which according to him were '*provoked by girls [sic]. . .they were too noisy and not careful*'.

**c) Family.** Two women who had family back at home reported it to be their 'biggest motivation' to stay healthy and to seek regular testing as they had to support them financially.

*I help my mother; my daughter is in a nice school here [in the Netherlands]. I know why I am doing this job and in order to keep doing it in future, I need to stay healthy.* (Female sex worker)

**Interviewer:** *Why do you test?*

**Respondent:** *Because I take care of myself, I am still working, I am 37. I help my family, I have a daughter she is 19 years old. She is in the University. She is not like me. She still does*

*not have a boyfriend and I started dating when I was 13 years old. . . back then, no sex. I told her all about men. . . all they want is to fuck. So she is careful and studies hard. She was studying hard always, reading books, she is clever. My daughter wants me to be happy. My daughter doesn't know what I do for a living. I told her, that my husband is rich. I also have three sisters whom I support; they all live in Ukraine. . .and their kids. . . like if they want an iPhone, I buy.* (Female sex worker)

## Meso-level factors

**a) Sex-work venue.** While some stakeholders and the owner of the escort agency believed that the income level facilitated the uptake of health services, female sex workers and healthcare providers did not confirm that. They rather associated income with a type of sex-work venue, which indeed facilitated access to health services. Thus, for women working indoors at a licensed sex-work venue, it was easier to access health services, as social nurses came to them. Local law facilitates access for healthcare providers to all licensed sex-work venues, but this is not possible in case of unlicensed ones, including individual work on the street or at home and online sex-work venues.

The majority of stakeholders reported sex workers working online to be the hardest group to reach. There are several organizations in Amsterdam doing online outreach, which implies: 1) weekly monitoring of several online platforms, where sex workers place their advertisements; 2) collecting information into the database (age, name, working hours, etc.); 3) checking license number; 4) evaluating risks based on a type of services provided, age and opening hours; 5) sending typical messages to all the numbers collected weekly. This approach was reported to be time-consuming while its effectiveness was unknown as there were seldom any responses. Moreover, it was impossible to examine who received the messages as escort agencies might use a third person (an operator) to communicate with clients. This strategy was criticised by one of the respondents because of the lack of individual approach:

> **Respondent:** *Most people don't respond to the text messages. . . [. . .] and they think, 'hey, this is just a standard text. . .I don't respond, just like a newsletter' or. . .and the others think 'well, I don't know what is this, I don't trust it'. You don't see any picture, you don't see any face, you don't. . .any address. Well, if you see pictures of the building, of the faces, it can make a different impression about the atmosphere. How they treat other sex workers.* (Sex worker, activist, employee of self-organized sex workers' organization)

**b) Social support.** Pimps, colleagues and peers could be both a source of support and a barrier to accessing health services, including HIV testing. For example, the owner of the escort agency described how he provides health advice to women on tourist visas. He has developed a text message named 'Operation Moydodyr' (Soviet cartoon promoting personal hygiene rules), which contained advice on what sex workers should do before sexual intercourse *'to stay safe'*, including a mandatory shower for clients, visual examination (dirty clothes, bad breath, rash on the genitals, etc.). Clients who do not pass muster receive three warnings before being blacklisted. This has been seen as a responsibility of the manager/owner to check clients and to have a special system where they can be checked. Female sex workers also used online communication (online forum) to report cases of abuse or violence. For example, if someone was robbed by a client or an escort agency did not provide the agreed level of support, women would report this situation, including a photo or contact details or screen prints of the online conversations. For example, five women on the online forum reported that the owner of an escort agency did not provide support to a woman who was caught by the Dutch police and then deported from the Netherlands. It was also mentioned

that he blackmailed a woman as she started to work on her own. This opened public debates between the owner and the women on the online forum, with supporters on both sides. Peers also use this channel to provide some information about symptoms of STIs, how to prevent STIs, where to go to a doctor when travelling or how to self-medicate.

In addition, a woman who works in a club described how peers influenced her decision to test regularly: '*All in the club knows if you do not do that, no one will work with you*'.

Moreover, it was clear that most women who enter sex work for the first time seek help of third parties in arranging travel (visas, housing, and advertising). For example, two sex workers reported that they were assisted by a female friend, who was already as a sex worker. All other women interviewed reported being engaged through a manager or owner of an escort agency. Also, we saw that with time women became more familiar with the local police and how things work on the ground. Their command of Dutch improves, so they are willing to maximize their independence as well as earnings, and consequently, some decide to work on their own. Female sex workers and a third-party work closely together, but usually the owner of an escort agency would have full information about a woman, including her ID, home address, nude photos and sometimes a video, while a woman would have nothing except a mobile phone, fake name and a bank account. Thus, in most cases, a woman's greatest fear is that her involvement in sex work will be denounced to her family and friends.

**c) Trust.** Most stakeholders agreed that trust was important for female sex workers; some mentioned also *'feeling safe'* or *'safe place'*, which was closely connected with providing health services confidentially and/or anonymously. It was easier to build trusting relationships with female sex workers working at venues than with women working online. One social nurse reported that it takes several visits to a club for women to get used to her.

Language is also an important factor for building relationships of trust with healthcare providers or social workers. All female sex workers interviewed identified the ability to speak in their mother tongue as one of the reasons why they preferred to use health services in their home country.

> *I try to test every six months. I travel to Ukraine every six months and I try to go to my doctor. [. . .] I have melanoma and she knows me so well. . .and she knows what to do, what I need. I feel comfortable. . . I don't want anyone to know. At least, no one else. And in our country they understand our language. . .it is easier.* (Female sex worker)

Moreover, one woman reported that the absence of out-of-pocket payments and ability to choose your doctor reduces her trust in Dutch health care. In Ukraine, paying high out-of-pocket fees made her comfortable as it gave her a feeling of control over the situation as well as the assurance that she has chosen the best doctor (which correlated with high costs). It was also possible to go to another doctor if she was not satisfied with the first one. But in the Netherlands, it was much stricter and doctors all follow the same protocol.

> *You know they [Dutch health workers] will always give you here paracetamol, if you have pain in the knee or something else, they give paracetamol. They act strictly accordingly to the book and I don't have a trustful contact with them.* (Female sex worker)

**d) Stigma and discrimination.** Issues of the stigma attached to sex workers and migrants were mentioned by most participants. For example, one participant said that nowadays even Dutch sex workers prefer not to go to a family doctor because of the persistent social stigma facing sex workers. Moreover, some participants connected greater stigma with perceptions in Dutch society, partly formed by the dominant media discourse that the majority of sex workers

are non-Dutch citizens, and victims of trafficking, who were coerced into sex work, and that their experience is horrific:

> *I spoke to sex workers. . . In early 90s, like. . . if they would be at somebody's birthday party they would tell people: 'Yes, I am a sex worker in the Red Light district'. And responses would generally be 'Yes!!??' Like people would be sort of excited like: 'How often do you get to meet a sex worker?' Like: 'It's really cool!!!'. . . People would ask, like: 'How is it? So, you must have had a lot of adventures and met so many interesting people! You must have had a lot of stories', and now, since like 2000s the responses have really changed. So, people really become, like: 'Oh my God! You must have seen a lot of horrible things! It must have been really terrible! How did you end up there?'. . .It's a very different attitude.* (Director, activist)

Thus, sex workers prefer to hide their identity or to go to specialized clinics, where they will not be treated as victims. Some participants also connected anti-migration movements, enlargement of the EU, the Dutch referendum on the Ukrainian Association agreement and implications of these 'attitudes' on policy change such as raising the age to work in the sex industry from 18 to 21, clients' responsibility to check the license/registration of sex-work venues, closing of 40% of all the brothels (192 of the previous 482 according to the Project 1012), etc.

Topics of the victimization of sex workers and placing all migrant sex workers under suspicion of having been trafficked are major issues for sex workers, as they are not allowed to self-identify. Thus, in most cases even when women do not recognize themselves as victims, they would be seen as such according to the law and the way this law is interpreted and applied in practice:

> *As I said before, even if they [migrant sex workers] don't feel as victims, they are victims to me. If they tell me stories like ok, somebody paid half-price for the ticket, somebody arranged this room, or hotel or half-price of it. 'I am sharing that [costs] with them [third parties], so I need to pay them. And how do they get the money from your sexual services? So, somebody is profiting from your work and do you know what the debt is? No, I don't know, but I am sure that, if I pay, etc. . .' So, for them [migrant sex workers] it is like a strategy to live and to survive and they don't even think that they are victims.* (Police officer)

Self-stigmatization of sex work along with a fear being recognized as a sex worker in her home country made women ashamed of who they are and to pretend to be of another nationality and avoid *'our people'*:

> *I'm ashamed when I hear Russian language; I pretend that I don't understand it. Luckily, I don't look like a Russian girl. . .All of them are stinkers; . . .dirty smelly stinkers. Who will pay extra money for such dirty stinkers? I am not a dirty stinker, I am not like them, and they are infected and ill.* (Female sex worker)

**e) Time and transport costs.**   All women interviewed as well as some stakeholders talked about the time and transport costs as a potential barrier to accessing HIV testing:

> *If you do not have an appointment you will not get medical services . . . And if you have your appointment at 14.10 and you are late. Once I was in a traffic jam, I was eight minutes late and they did not let me have my appointment. And when you are waiting for them it's ok.* (Female sex worker)

For example, some stakeholders mentioned how opening a new STI clinic outside Amsterdam's city centre or a mobile testing facility could facilitate HIV testing. Moreover, one stakeholder described a pilot project where a self-testing approach was used. Although there were some concerns regarding the link to treatment and HIV care, the ability to pay for self-testing kits, and possible misuse or misinterpretation of test results, some individuals appreciated this approach as it was fast, easy to perform and could be done without involving anyone else.

## Macro-level factors

**a) Sex-work policies vs anti-migrant movement.**   Opponents of the Dutch prostitution policy and law perceived it to be permitting growing demands to use HIV/STI testing services for non-legal female sex workers from EE and non-EU countries in the Netherlands. This is often associated with a trend in which policy and law in relation to prostitution become increasingly strict and therefore have increasingly repressive effects on non-legal migrant female sex workers from EE, non-EU countries. For example, one of the social nurses experienced the separation between legal and non-legal prostitution as a barrier to doing her work:

*I feel that the non-legal sector is much less easy to access than the legal sector. So, if the non-legal sector becomes legal, it would be much easier to access them [sex workers]. Prostitution will always exist, no matter if it is legal or non-legal, but the accessibility for healthcare providers becomes less if it is non-legal.* (Social nurse)

One participant questioned a definition of the voluntary nature of sex work and thought that many legal sex workers do not voluntarily choose this work, while the government pretends that they have freely chosen to be a sex worker:

*It is suggested that exploitation of women [who are SWs] can be eradicated if you only allow sex workers to work when they have self-chosen for prostitution. However, these measures are not solving anything. . .I believe that if a person wants to sell his or her own body, some kind of damage has probably occurred in the past. So, what is being meant with self-chosen prostitution?* (Healthcare provider)

**b) Contradicting tasks of government organizations.**   Most of the participants perceived the combination of providing health care and the government measures that aim to reduce non-legal prostitution as a barrier to sex workers' trust in government organizations in the Netherlands. For example, the programme manager prostitution stated that this combination is not always easy to apply:

*In a situation where a healthcare provider, who has a confidential relationship with a sex worker, is confronted with human trafficking of that sex worker, the connection with the police needs to be made. However, this would damage the confidential relationship with that sex worker; we do not want that.* (Programme manager prostitution)

The physician pointed to the discouraging effects of the municipal prostitution policy to focus on non-legal migrant female sex workers from EE, non-EU countries:

*If you are not being supported by your own municipality, it feels like a barrier to focus our efforts more on non-legal sex workers. . .that is actually really weird from a public health perspective.* (Physician)

The programme manager prostitution mentioned that every sex worker should have similar access to take an HIV/STI test, but not to other things, such as Dutch language classes. However, the allocation of financial resources also determined the focus of the municipal health services:

*Our core business is the legal sector because that is being financed. [. . .] Time–we do not have enough time to reach out to escort ladies.* (Programme manager prostitution)

**c) Free antiretroviral therapy.** Although human rights aspects, such as universal access to comprehensive prevention programmes, treatment, care and support, constitute a fundamental principle and were introduced in the Netherlands long ago, some stakeholders as well as all female sex workers interviewed had fragmented knowledge about whether and how ART treatment would be covered for non-EU female sex workers. To address this issue, one NGO has developed various educational materials and training aimed at health professionals. One NGO representative described a specific case of a Ukrainian woman who was pregnant and was diagnosed HIV-positive, and all the difficulties they faced in helping her with beginning ART, as she also had other health problems, which involved additional costs (to cover the necessary analysis, tests, and visits to health specialists). Even though the NGO managed to provide some support, which took several months of hard work, at some point the woman returned to Ukraine:

*She could access [treatment] as an exceptional case, we did it, we found a specialist, because she needed more than just ART. [. . .] She needed different specialists. . . Okay, finally we were able to find the supporting organization, that did more or less what we asked to do, but at some point we lost contact with her. We heard from her mother [. . .] the grandmother of the child, that she moved to Israel for work.* (Director, NGO)

## Discussion

This paper sought to examine the context-specific vulnerabilities of non-EU migrant female sex workers and to describe their experiences and perceptions of existing HIV testing services in Amsterdam. In summary, the analysis highlights six barriers to HIV testing most commonly reported by research participants: 1) migration and sex-work policies; 2) stigma, including self-stigmatization; 3) lack of trust in healthcare providers or social workers; 4) low levels of Dutch or English languages; 5) negative experience in accessing healthcare services in the home country; and 6) low perceived risk and HIV-related knowledge. Having a family and children, social support and working at the licensed sex-work venues could facilitate HIV testing among migrant female sex workers in Amsterdam. Moreover, this research adds the voices of EE, non-EU women involved in escort services, which often remain inaudible to research, health providers and the public [1].

As reported, structural factors and conditions, such as sex work and migration polices, and sex workers' immigration status, are key predictors of vulnerability [28, 39]. Despite the fact that sex work is legalized in the Netherlands, migrant non-EU female sex workers are not eligible to apply for a work permit to be employed as sex workers [7]. Thus, women either have to find ways to become eligible or they try to hide their work and remain 'invisible' in the country. Moreover, the contradictory tasks of government organizations, which involve a combination of providing health care and various control measures, may further limit the accessibility of health services for migrant female sex workers. Over the last years, regulations on sex work in Amsterdam and across the Netherlands have been fostered (Project 1012, raising of the age

for sex-work employment and clients' responsibility, etc.). It is still unclear how a licensing system would work for individuals who occasionally practise sex work at home or online, provide a 'girlfriend experience' or go on paid dates; nor whether the licensing approach, with its comprehensive system of regulations, actually increases security. It does, however, raise a number of concerns: 1) high administrative costs needed to cover regular check-ups of licensed sex-work venues; 2) poor quality of health services provided onsite, lack of confidentiality and insensitive treatment of sex workers; and 3) not being effective for unlicensed sex-work venues [28]. Moreover, there are sound arguments against formally involving governments in the organization of sex work through licensing or registration [40]. On the other hand, the decriminalization of sex work may bring the greatest benefit to sex workers and broader community, provided that effective laws against sexual abuse, exploitation, coercion and trafficking of minors and adults are in place [28].

While working at a licensed sex-work venue could promote the uptake of health services, including HIV testing, unlicensed venues such as online escort agencies may impede it. This should be considered as dual process: from one perspective, the unlicensed sector is difficult to reach and is not considered to be a priority for local healthcare providers as there is no rigorous data proving that there is such a need. From a different perspective, sex workers in the unlicensed sector may avoid any contact with officialdom, including health workers. Local health facilities employ the paradigm of licensed windows, and brothels located in a specific area, like the Red Light districts. So, they can enumerate all the venues, ask for their registration number from the City Council and then, safeguarded by the sex-work policy, knock on their door to gain physical access to sex workers. However, very little attention is paid to the online sex-work agencies and individuals working without a license. This may be the consequence of a municipal prostitution policy and allocation of resources, but also of the absence of rigorous evidence regarding size of the unlicensed sex-work sector, including online-based sex work, and the difficulty in reaching female sex workers using the internet. There is limited evidence on how migrant sex work operates through numerous online platforms and social media [1], although this form of sex work is likely to increase in the future.

Another important trend is that most Dutch policies on sex work were initially mapped onto street-based and later licensed sex-work spaces, while the internet was a 'legal geography of exclusion' that made such sex workers invisible to the state and the law[6]. Moreover, since publishing of a government action plan (Ordening bescherming van de prostitutiesector) in 2004, regular check-ups and 'supervision' of licensed sex-work spaces have become in the focus of municipal legislation in the Netherlands. Consequently, unlicensed sex-work spaces fell below the radar as they were classified as criminalized forms that cannot be incorporated with a legal framework. We call for a broader understanding of sex-work identities, practices and lived experiences, suggesting that any given female sex worker could combine multiple geographies of sex work (based on migration status, legal definition of sex-work space, licensing system, continuity and regularity of engagement in sex work) throughout her life.

This study also highlights that having a relationship of trust with healthcare providers can facilitate HIV testing. In addition, we observed how trust is related to other factors (e.g. knowledge of local policies or fear of being recognized as sex workers) and may be affected by negative background experience in the home country or low levels of Dutch or English languages. The building of trust can be described as both a process in the establishment of a caring relationship between the healthcare provider and the patient and an outcome of that relationship [41]. This involves the ability to relate to each other and to be mutually interested in having a meaningful social relationship[42, 43]. In this respect we suggest considering several factors: the possibility of communicating in the patient's native language, knowledge of local policies and how they work, confidentiality or even anonymity (for those violating migration or sex-

work employment policies), and the absence of stigma and discriminatory attitudes towards sex workers, PLHIV and migrants. In addition, profound knowledge of the culture of internet-based sex workers, including their self-definitions, may play a crucial role in designing strategies which aim to build trusting relationships with these sex workers. For example, women providing escort services, 'girlfriend experience' or doing paid or car dates may not associate themselves with health services designed for prostitutes or sex workers, and so may not seek them.

We have described experiences of internet-based escort-agency female sex workers, who travelled as tourists across several EU countries, and women who had valid residence and work permit and worked in a club. In both cases, however, women preferred not to report their sex work to doctors, friends, neighbours and their family. In line with previous studies, we found that the stigma associated with sex work, including self-stigmatization and HIV- and AIDS-related stigma, is a barrier to accessing health services, including HIV testing. We suggest considering both sex work and HIV- and AIDS-related stigma not as a matter of individual processes, experiences or perceptions, but rather as a social process linked to power, inequality and exclusion [44, 45]. This social process is inherently linked to the production and reproduction of structural inequalities, and should be understood as a part of the political economy of social exclusion of individuals with 'undesirable differences' or 'deviances' ('dislike of the unlike'), who are the objects of subordination in the modern world [45]. So, in order to combat stigma, we have to think about structural and environmental interventions aimed at transforming the local contexts in which migrant sex workers face double or even triple stigma. While community mobilization, empowerment, advocacy, and a rights-based approach were among most common strategies used to address sex work and HIV- and AIDS-related stigmas and promote the uptake of services [46–48], it is not clear if the same strategies would be as effective with highly mobile migrant sex workers. Nevertheless, we believe that existing local networks, unions and community-led organizations of sex workers in their home countries, especially in case of circular migration of EE, non-EU migrant female sex workers across multiple EU countries, might serve as a potential bridge between women and health professionals in destination countries. Moreover, networks, unions and organizations of sex workers, as well as organizations representing migrant sex workers, should be included in the policy process regarding sex work, migration and trafficking issues.

Similar to the results reported in the literature, we observed that the social support of peers, family, friends and escort agency owners may assist women in accessing health services [46, 49–56]. Moreover, having children may increase women's self-perceived risk and facilitate regular health checks, or may also be explained by women's and particularly pregnant women's greater exposure to HIV testing, which is generally offered in reproductive and antenatal care settings [57–59]. Thus, we described situations when migrant female sex workers on 'tours' rely on information they receive from peers or agency owners. Moreover, self-medication and travelling back to their home country were most common health-seeking strategies. These involve questions of language, trust and perceived control over the situation, which should be explored in future studies. In this situation, we suggest considering self-testing [51, 60] and promoting it online in close cooperation with EE sex-work activists, leaders and networks, in ways that are consistent with cultural traditions. Moreover, we call for considering migration as a dynamic process, which occurs across the home country's borders to multiple destination countries and could be repeated many times, and in various ways [61]. Thus, future research should address factors mediating access to health services across different stages of the migration cycles of EE, non-EU migrant female sex workers.

## Strengths and limitations

A strength of this study is that it relies on real-life evidence and everyday practices of three major actors: 1) key stakeholders, including health professionals, representatives of NGOs and sex worker-led organizations, the police and other municipal functionaries;, 2) a manager of escort agency and sex workers' clients; and 3) migrant female sex workers. As with all studies that collect self-reported data, there was a risk of respondents giving socially desirable answers. In order to minimize this risk, taking account of the sensitivity of the research topic, we reassured all participants that any information they provide would remain confidential; if participants preferred, their voice was not recorded.

A major limitation is the small number of migrant female sex workers interviewed. Even though we cooperated with local NGOs, we could obtain access only women living and working legally in Amsterdam. Thus, we had to contact an escort agency and gain its trust, which was a slow and time-consuming process. Due to the small number of migrant female sex workers interviewed, we are uncertain whether data saturation has been realized.

When using the term 'Russian speaking', we do not suggest that we ignored differences between the cultures and national identities of Belarus, Moldova, Russia and Ukraine. In this respect, we mean simply the ability to speak and understand Russian, and refer to the shared history of the Soviet Union.

## Conclusion

EE, non-EU migrant female sex workers in Amsterdam face multiple barriers to accessing health services, including HIV testing. It is important to acknowledge women's differentiated and diverse experiences, shaped by intrapersonal, interpersonal, community, network and policy-level factors, with stigma being at the core. We call for scaling up outreach interventions that focus on female sex workers and migrant female sex workers working online, who remain an invisible part of the sex-work industry. Having relationships of trust with women sex workers, and providing social support, may facilitate HIV testing among migrants. In this regard, it is important that networks, unions and organizations of sex workers and migrant sex workers play a fundamental role in the formulation of sex work and migration policies as well as programme delivery. We also encourage interventions involving cultural mediators and local NGOs, and sex-worker-led organizations in women's home countries. Future studies may usefully examine new sex-work geographies among EE, non-EU migrant sex workers.

## Supporting information

**S1 Appendix. Annex 1: Data extraction form for analysis of the online advertisements EE migrant FSWs in Amsterdam.**
(DOCX)

**S2 Appendix. Annex 2: Interview guide for key stakeholders (Phase 1).**
(DOCX)

**S3 Appendix. Annex 3: In-depth Interviews with the EE, non-EU migrant FSWs (Phase 2).**
(DOCX)

**S4 Appendix. Annex 4: In-depth Interviews with stakeholders (Phase 3).**
(DOCX)

**S5 Appendix. Annex 5: Coding framework.**
(DOCX)

## Acknowledgments

We are grateful to the people who participated for sharing their stories. We are thankful to Mrs Deborah Eade for English language editing. We are thankful to Maria Roura for supporting the idea of this study.

## Author Contributions

**Conceptualization:** Anna Tokar, Jacqueline E. W. Broerse.

**Data curation:** Anna Tokar, Robbert Hengeveld.

**Formal analysis:** Anna Tokar, Jacob Osborne, Robbert Hengeveld.

**Funding acquisition:** Anna Tokar.

**Investigation:** Anna Tokar, Robbert Hengeveld.

**Methodology:** Anna Tokar, Robbert Hengeveld, Jacqueline E. W. Broerse.

**Project administration:** Anna Tokar, Jacqueline E. W. Broerse.

**Resources:** Anna Tokar, Jacqueline E. W. Broerse.

**Software:** Anna Tokar.

**Supervision:** Jeffrey V. Lazarus, Jacqueline E. W. Broerse.

**Validation:** Anna Tokar, Jacob Osborne.

**Visualization:** Anna Tokar.

**Writing – original draft:** Anna Tokar.

**Writing – review & editing:** Anna Tokar, Jacob Osborne, Robbert Hengeveld, Jeffrey V. Lazarus, Jacqueline E. W. Broerse.

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
