## [Decision Letter · Decision Letter 0]

6 Jan 2020

PONE-D-19-24622

‘I don’t want anyone to know’: experiences of obtaining access to HIV testing by Eastern European non-European Union sex workers in Amsterdam

PLOS ONE

Dear Mrs Tokar,

Thank you for submitting your manuscript to PLOS ONE. After careful consideration, we feel that it has merit but does not fully meet PLOS ONE’s publication criteria as it currently stands. Therefore, we invite you to submit a revised version of the manuscript that addresses the points raised during the review process.

Please, improve the paper according to the referee comments (see below).

We would appreciate receiving your revised manuscript by Feb 17 2020 11:59PM. To enhance the reproducibility of your results, we recommend that if applicable you deposit your laboratory protocols in protocols.io, where a protocol can be assigned its own identifier (DOI) such that it can be cited independently in the future. For instructions see: http://journals.plos.org/plosone/s/submission-guidelines#loc-laboratory-protocols

We look forward to receiving your revised manuscript.

Kind regards,

Joan A Caylà, PhD, MD

Academic Editor

PLOS ONE

Journal Requirements:

2. Please revise your ethics statement in the submission form to reflect the information in sec. 2.4.

3. Please can you udpate the corresponding authors contact details to provide an institutional email address.

4. We noticed you have some minor occurrence of overlapping text with the following previous publication, which needs to be addressed: Tokar, Anna, et al. "HIV testing and counseling among female sex workers: a systematic literature review." AIDS and Behavior 22.8 (2018): 2435-2457. In your revision ensure you cite all your sources (including your own works), and quote or rephrase any duplicated text outside the methods section. Further consideration is dependent on these concerns being addressed.

5. We note that Figures 1 and 2 in your submission contain copyrighted images. All PLOS content is published under the Creative Commons Attribution License (CC BY 4.0), which means that the manuscript, images, and Supporting Information files will be freely available online, and any third party is permitted to access, download, copy, distribute, and use these materials in any way, even commercially, with proper attribution. For more information, see our copyright guidelines: http://journals.plos.org/plosone/s/licenses-and-copyright.

1.         You may seek permission from the original copyright holder of Figure(s) [#] to publish the content specifically under the CC BY 4.0 license.

6. Please do not include funding sources in the Acknowledgments or anywhere else in the manuscript file. Funding information should only be entered in the financial disclosure section of the submission system. https://journals.plos.org/plosone/s/submission-guidelines#loc-acknowledgments.

7. We note that you have indicated that data from this study are available upon request. PLOS only allows data to be available upon request if there are legal or ethical restrictions on sharing data publicly. For information on unacceptable data access restrictions, please see http://journals.plos.org/plosone/s/data-availability#loc-unacceptable-data-access-restrictions.

Reviewers' comments:

Reviewer's Responses to Questions

**Comments to the Author**

1. Is the manuscript technically sound, and do the data support the conclusions?

Reviewer #1: Yes

2. Has the statistical analysis been performed appropriately and rigorously? 

Reviewer #1: Yes

3. Have the authors made all data underlying the findings in their manuscript fully available?

Reviewer #1: Yes

4. Is the manuscript presented in an intelligible fashion and written in standard English?

Reviewer #1: Yes

5. Review Comments to the Author

Reviewer #1: This is an original publication of qualitative data exploring barriers and facilitators to HIV testing (as a proxy for a wider range of HIV services) available to migrant sex workers in the Netherlands. The motivation for the study is policy-oriented rather than an acute crisis of rising HIV infections or a high HIV mortality among the population of interest, although evidence of a high lack of awareness of HIV infection among sex workers is probably the biggest "signal" presented by authors. The contradictory policies of legalized sex work versus restrictive employment policies for immigrants have created a "loophole" of sorts - services can't be accessed by those who may need them the most.

The introduction is long and reads more like a policy or social science paper in some ways. However, because the observations and conclusions are unique to an often forgotten, marginalized group, with little opportunity to voice their experiences (as evidenced by the very limited number of sex workers that could be interviewed), we must recognize the value of these data as a baseline, a platform to grow the body of research on this subject. It would have been interesting to include questions to the stakeholders on how best to access migrant sex workers operating "under the radar."

I commend the authors on a well-written, well-reasoned manuscript on a fairly novel, under-explored area of research, especially given the evidence that incidence is declining in most parts of the world, except for Eastern Europe where may of the migrant sex workers originate.

Minor revisions requested:

1. Recommend "sex workers who use drugs" rather than "drug-using sex workers."

2. Check wording please: "...problems of being [able] to work legally..." - consider more direct wording. Rather than problems of being able to, consider "...together with being unable to work legally..."

3. Thank you for highlighting the limited numbers in phase 2 - migrant sex workers, particularly those working without legal status, are absent from data collection. Could this be expanded? I think it's a methodological note worth explaining - so that future research on this area/this population can learn from the authors' experiences. What would they have done differently if they could re-design their strategy to access this population?

4. Should phase 3, interviewee number 6 be re-categorized to phase 2?

5. Is there a reason that only one interview tools was included in the submission? "Interview guide for key stakeholders" - is this from phase 1 or phase 3?

6. Were the selected criteria for "key stakeholders" in phases 1 and 3 different? It seems like the same pool of stakeholders?

7. To readers less familiar with qualitative methods, please ensure the difference between semi-structured interview and in-depth interview and the reasons to select one method or the other are clear.

8. Wording check: "...and use their power [to] get extra sexual services for free.

6. PLOS authors have the option to publish the peer review history of their article (what does this mean?). If published, this will include your full peer review and any attached files.

Reviewer #1: No

---

## [Author Response · Author response to Decision Letter 0]

27 Apr 2020

February 24, 2020

Academic Editor

PLOS ONE

Dear Prof. Joan A.Caylà

On behalf of my co-authors (J. Osborne, R. Hengeveld, Jeffrey V. Lazarus, Jacqueline E.W. Broerse) I am re- submitting our manuscript entitled “I don’t want anyone to know’: experiences of obtaining access to HIV testing by Eastern European non-European Union sex workers in Amsterdam” (Manuscript ID: PONE-D-19-24622). Below, we provide a response to each point raised by the academic editor and reviewer(s).

Comments Responses

Journal Requirements:

1.Please ensure that your manuscript meets PLOS ONE's style requirements, including those for file naming. We have corrected the file naming to meet PLOS ONE's style requirements.

2.Please revise your ethics statement in the submission form to reflect the information in sec. 2.4. We have corrected ethics statement in the submission form.

3.Please can you update the corresponding authors contact details to provide an institutional email address. We have updated the corresponding authors contact details and provided an institutional email address.

(Page 1)

4.We noticed you have some minor occurrence of overlapping text with the following previous publication, which needs to be addressed: Tokar, Anna, et al. "HIV testing and counseling among female sex workers: a systematic literature review." AIDS and Behavior 22.8 (2018): 2435-2457. In your revision ensure you cite all your sources (including your own works), and quote or rephrase any duplicated text outside the methods section. Further consideration is dependent on these concerns being addressed. Thank you very much for the comment.

We have added this reference.

(MethodsSection-Page 11, reference [36]

Figure 2-Page 12 reference [36])

5.We note that Figures 1 and 2 in your submission contain copyrighted images. All PLOS content is published under the Creative Commons Attribution License (CC BY 4.0), which means that the manuscript, images, and Supporting Information files will be freely available online, and any third party is permitted to access, download, copy, distribute, and use these materials in any way, even commercially, with proper attribution. Both figures were developed by the authors. 

Figure 1 represents the project phases of the current studies and was exclusively developed for this publication by the authors. 

(Page 8)

Figure 2 was developed as a result of the systematic review, which was published in the AIDS and Behavior (Tokar A, Broerse JEW, Blanchard J, Roura M. HIV Testing and Counseling Among Female Sex Workers: A Systematic Literature Review. (2018). AIDS and Behaviour. Vol.13:(0123456789), [Original research].https://doi.org/10.1007/s10461-018-2043-3).

We have added reference under Figure 2.

(Page 12, reference [36] )

6.Please do not include funding sources in the Acknowledgments or anywhere else in the manuscript file. Funding information should only be entered in the financial disclosure section of the submission system. We have deleted funding information from Acknowledgments. It is only included in the financial disclosure section of the submission system and funding section of the manuscript.

(Funding -Page 30)

7.We note that you have indicated that data from this study are available upon request. PLOS only allows data to be available upon request if there are legal or ethical restrictions on sharing data publicly. 

b) If there are no restrictions, please upload the minimal anonymized data set necessary to replicate your study findings as either Supporting Information files or to a stable, public repository and provide us with the relevant URLs, DOIs, or accession numbers. We would like to keep the data available upon request because of several reasons:

a) de-identified transcripts of interviews with stakeholders might still allow to identify who those persons were as they describe their work routine. Moreover, there are a limited number of organizations focusing on sex workers or migrant sex workers in Amsterdam (including unique institutions, e.i., The National Rapporteur on Trafficking in Human Beings and Sexual Violence against Children or special police departments), which makes it easier to identify people. 

b) The interviews with migrant sex workers contain sensitive information, including information about women’s health and family status, as well as detailed descriptions of how they violated some work and migration policies either in past or in present.

c) Collected advertisements, contain personal information (contact details, nationality and personal photos), and even after deleting all personal information it might be possible to use some parts of the text in order tofind these advertisements online as they are still available at the website of the escort agency.

Review Comments to the Author

1. Recommend "sex workers who use drugs" rather than "drug-using sex workers." Thank you very much for the comment. We have changed"drug-using sex workers" to"sex workers who use drugs".

(Introduction Section- Page 3)

2. Check wording please: "...problems of being [able] to work legally..." - consider more direct wording. Rather than problems of being able to, consider "...together with being unable to work legally..." We have changed "...problems of being [able] to work legally... " to "...together with being unable to work legally..."

(Introduction Section-Page 4)

3. Thank you for highlighting the limited numbers in phase 2 - migrant sex workers, particularly those working without legal status, are absent from data collection. Could this be expanded? I think it's a methodological note worth explaining - so that future research on this area/this population can learn from the authors' experiences. What would they have done differently if they could re-design their strategy to access this population? Thank you very much for the comment.

We have written a separate methodological paper focusing on different methodological issues, challenges as well as presenting some reflections. This paper is now under review. We have added: 

“A more detailed descriptions and reflections on methodological aspects of this study, including sampling, data collection and ethical issues, could be found in another article, which we have written.”

(Methods Section- Page13)

4. Should phase 3, interviewee number 6 be re-categorized to phase 2? Interviewee number six, was not a migrant from Eastern Europe. This was a Dutch resident, who was born in the Netherlands and at the moment of our interview, worked as a street-based sex worker. Also, this person was an activist and a member of one sex worker-led organization in the Netherlands. 

5. Is there a reason that only one interview tools was included in the submission? "Interview guide for key stakeholders" - is this from phase 1 or phase 3? The initially included guide was from Phase 1.

We have included now all the guides as annexes.

(Annex 2-4 - Pages 35-44)

6. Were the selected criteria for "key stakeholders" in phases 1 and 3 different? It seems like the same pool of stakeholders? The pool of respondent for Phase 3 was defined based on the results of Phases 1 and 2. It included some of the NGOs, which we have contacted in the Phase 1 (three of them), yet we have never talked twice to the same person. Moreover, as you may see from the list of participants (Table 1), in Phase 1 a broad range of stakeholders was included (e.g., journalists), while in Phase 2 we were able to focus exclusively on those stakeholders working with EE non-EU migrant FSWs.

7. To readers less familiar with qualitative methods, please ensure the difference between semi-structured interview and in-depth interview and the reasons to select one method or the other are clear Thank you very much for the comment. We have added a description of both methods as well as reasons why these methods have been applied. Terms “semi-structured interviews” and “in-depth, narrative interviews” were used in accordance with definitions given by Judith Green and Nicki Thorogood, “Qualitative methods for Health Research”, SAGE Ltd, 2004. 

“Since we aimed to collect information on predefined topics in this first phase of the project, we conducted semi-structured interviews, in which a researcher set the agenda in terms of topics covered, but the respondent determined the kinds of information obtained as well as the relative importance of this information.” (Methods Section – Pages 8-9)

“In Phase 2 we used in-depth, narrative interviews as this method allowed respondents to have enough time to develop their own accounts of the issues important to them, while interviewer served rather as a facilitator of this process. Respondents determined the conversation flow.”

(Methods Section – Pages 9)

8.Wording check: "...and use their power [to] get extra sexual services for free Thank you very much for the comment.

We have changed “and use their power to get extra sexual services for free.”

(Results Section– Page 15)

We also would like to mention that we have initially applied to receive a publication fee waiver. Ms. Tokar, a first author, a Ukrainian, is a Trans Global Health PhD student (ERASMUS MUNDUS 2009-2013, EMJD, Action 1B, Agreement No. 2013-685 0039; http://www.transglobalhealth.org).ISGlobal, University of Barcelona, Barcelona, Spain is her host institute. Ms. Tokar’s living allowances and cost of her PhD research (as bench fees) are funded by the European Commission (EC). Under the PhD scheme, Ms. Tokar received a limited amount of money as bench fees (around 4,000€/year for her 3-year PhD research project) and there is no fund dedicated to cover publishing costs. In the Erasmus Mundus Program Guide on page 43-45 you can find all the information about the grant award. Moreover, currently Ms. Anna Tokar does not receive a salary, as she has finished her PhD and now is about to defend her thesis.

Considering Ms. Tokar’s research funding situation and importance of her research topic, we ask to consider her application for receiving a publication feewaiver (partial or full) from the PLOS ONE.

Thank you very much for your consideration of this manuscript.

Sincerely, 

Anna Tokar

Barcelona Institute for Global Health (ISGlobal),

Hospital Clínic – University of Barcelona

Rosselló, 132, 4th

ES-08036, Barcelona, Spain

---

## [Decision Letter · Decision Letter 1]

29 May 2020

I don’t want anyone to know’: experiences of obtaining access toHIV testing by Eastern European,non-European Union sex workers in Amsterdam, the Netherlands.

PONE-D-19-24622R1

Dear Dr. Tokar,

We are pleased to inform you that your manuscript has been judged scientifically suitable for publication and will be formally accepted for publication once it complies with all outstanding technical requirements.

With kind regards,

Joan A Caylà, PhD, MD

Academic Editor

PLOS ONE

Additional Editor Comments (optional):

Reviewers' comments:

Reviewer's Responses to Questions

**Comments to the Author**

1. If the authors have adequately addressed your comments raised in a previous round of review and you feel that this manuscript is now acceptable for publication, you may indicate that here to bypass the “Comments to the Author” section, enter your conflict of interest statement in the “Confidential to Editor” section, and submit your "Accept" recommendation.

Reviewer #1: All comments have been addressed

2. Is the manuscript technically sound, and do the data support the conclusions?

Reviewer #1: Yes

3. Has the statistical analysis been performed appropriately and rigorously? 

Reviewer #1: Yes

4. Have the authors made all data underlying the findings in their manuscript fully available?

Reviewer #1: Yes

5. Is the manuscript presented in an intelligible fashion and written in standard English?

Reviewer #1: Yes

6. Review Comments to the Author

Reviewer #1: (No Response)

7. PLOS authors have the option to publish the peer review history of their article (what does this mean?). If published, this will include your full peer review and any attached files.

Reviewer #1: No

---

## [Editor Report · Acceptance letter]

12 Jun 2020

PONE-D-19-24622R1 

‘I don’t want anyone to know’: experiences of obtaining access to HIV testing by Eastern European, non-European Union sex workers in Amsterdam, the Netherlands 

Dear Dr. Tokar:

I'm pleased to inform you that your manuscript has been deemed suitable for publication in PLOS ONE. Congratulations! Your manuscript is now with our production department. 

Kind regards, 

on behalf of

Professor Joan A Caylà 

Academic Editor

PLOS ONE